# Ethylene: A Master Regulator of Salinity Stress Tolerance in Plants

**DOI:** 10.3390/biom10060959

**Published:** 2020-06-25

**Authors:** Riyazuddin Riyazuddin, Radhika Verma, Kalpita Singh, Nisha Nisha, Monika Keisham, Kaushal Kumar Bhati, Sun Tae Kim, Ravi Gupta

**Affiliations:** 1Department of Plant Biology, Faculty of Science and Informatics, University of Szeged, Közép fasor 52, H-6726 Szeged, Hungary; riyazkhan24992@gmail.com; 2Doctoral School in Biology, Faculty of Science and Informatics, University of Szeged, H-6720 Szeged, Hungary; 3Department of Biotechnology, Visva-Bharati Central University, Santiniketan, West Bengal 731235, India; radhikaverma040@gmail.com; 4School of Biotechnology, Gautam Buddha University, Greater Noida, Uttar Pradesh 201312, India; kalpita.singh4@gmail.com; 5Department of Integrated Plant Protection, Plant Protection Institute, Faculty of Horticultural Sciences, Szent István University, Páter Károly utca 1, H-2100 Gödöllo, Hungary; nisha3005n@gmail.com; 6Department of Botany, University of Delhi, New Delhi 110007, India; monik.kim@gmail.com; 7Louvain Institute of Biomolecular Science, Catholic University of Louvain, B-1348 Louvain-la-Neuve, Belgium; kaushalkbhati@gmail.com; 8Department of Plant Bioscience, Pusan National University, Miryang 50463, Korea; 9Department of Botany, School of Chemical and Life Sciences, Jamia Hamdard, Hamdard Nagar, New Delhi 110062, India

**Keywords:** ROS, ethylene, antioxidants, salinity stress, photosynthesis, programmed cell death, seed germination, hormone cross-talk

## Abstract

Salinity stress is one of the major threats to agricultural productivity across the globe. Research in the past three decades, therefore, has focused on analyzing the effects of salinity stress on the plants. Evidence gathered over the years supports the role of ethylene as a key regulator of salinity stress tolerance in plants. This gaseous plant hormone regulates many vital cellular processes starting from seed germination to photosynthesis for maintaining the plants’ growth and yield under salinity stress. Ethylene modulates salinity stress responses largely via maintaining the homeostasis of Na^+^/K^+^, nutrients, and reactive oxygen species (ROS) by inducing antioxidant defense in addition to elevating the assimilation of nitrates and sulfates. Moreover, a cross-talk of ethylene signaling with other phytohormones has also been observed, which collectively regulate the salinity stress responses in plants. The present review provides a comprehensive update on the prospects of ethylene signaling and its cross-talk with other phytohormones to regulate salinity stress tolerance in plants.

## 1. Introduction

Ethylene, the first gaseous plant hormone to be identified, is a key regulator of plant growth and development. Although growth-regulating effects of ethylene were first observed in 1901 by a Russian physiologist Dimitry K. Neljubov, ethylene was established as a plant hormone almost 60 years later in 1965 [1]. The biosynthetic pathway of ethylene production was elucidated in 1970–80, while its signaling components were identified in the 1990s after the development of *Arabidopsis thaliana* as a model system that facilitated the development and screening of genetic mutants [2]. Subsequent studies on ethylene led to the identification of an array of genes and transcriptional factors working downstream that widen our understanding of how this colorless and odorless plant hormone functions [3]. Because of the gaseous nature of ethylene, it can easily diffuse to nearby cells and, therefore, ethylene production predominantly takes place locally at the site of its action. Earlier work showed that ethylene biosynthesis is dramatically induced during fruit ripening and leaf senescence, among others [4]. Ethylene is also well known for its triple response which includes (i) inhibition of hypocotyl and root elongation, (ii) swelling of the hypocotyl, and (iii) exaggerated tightening of the apical hook [5].

In addition to regulating plant growth and development, research in the past two decades has also highlighted the involvement of ethylene in regulating plant responses to various biotic and abiotic stresses [6,7,8,9]. Among different abiotic stresses, ethylene has emerged as one of the important positive mediators for salinity stress tolerance in the model plant *A. thaliana* as well as in many crop plants including grapevines, maize, and tomato [10,11,12,13]. Salinity stress is one of the major abiotic stresses, posing a major threat to agricultural productivity [14,15]. Globally, more than 20% of irrigated land is affected by salinity stress, resulting in an average yield fall of more than 50% for major crops [16,17]. As per the Food and Agriculture Organization (FAO), 13,003 million hectares (Mha) of the global land area is devoted to agriculture [18] in which soil salinity has adversely affected about 30% of the irrigated land and 6% of the total land area [19,20], resulting in a monetary loss of approximately 27.3 billion USD per year [21]. Plants growing in geographically over-salted soils in areas where hydraulic lifting of saline underground water occurs and in coastal areas are frequently exposed to salinity stress [22]. The decline of farmable land due to salinity is a major concern for food security in the current and future scenario of a growing population which has been estimated to reach 8.5 billion over the next 25 years [23]. Salinity stress is exerted because of excess of one or more salt ions in the soil including sodium (Na^+^), bicarbonate (HCO_3_^−^), magnesium (Mg^2+^), sulfate (SO_4_^−2^), potassium (K^+^), chloride (Cl^−^), calcium (Ca^2+^), and carbonate (CO_3_^−2^) [24]. Salinity stress negatively influences seed germination, growth, physiology, productivity, and reproduction and sometimes even results in death under severe conditions [25]. At the onset of salinity stress, the capacity for water absorption by roots decreases and the transpiration rate increases due to osmotic imbalance, thereby generating hyperosmotic stress [26,27]. Osmotic stress, in turn, induces closure of stomata, which restricts CO_2_ uptake, resulting in reduced carbon fixation and assimilation in leaf tissues [28,29]. Rates of photosynthesis and carbohydrate production are therefore reduced, which impacts plant growth and yield. Other consequences of stomatal closure include reduced evapotranspiration of water and accumulation of reactive oxygen species (ROS) [30]. Salinity stress-induced ROS accumulation can lead to uncontrolled oxidation of membranes, proteins, and DNA, ultimately resulting in cell death [31]. In order to maintain ROS homeostasis [32], plants are equipped with a powerful and multifaceted antioxidant system, consisting of enzymatic and nonenzymatic components. While enzymatic components include superoxide dismutase (SOD), ascorbate peroxidase (APX), catalase (CAT), glutathione reductase (GR), guaiacol peroxidase (POD), glutathione transferase (GST), glutathione peroxidase-like (GPXLs), and thioredoxin peroxidase (TPOX), non-enzymatic antioxidants consist of ascorbic acid (AsA), glutathione (GSH), α-tocopherol (TOC), flavonoids, and carotenoids [33,34]. Successful detoxification of the stress-induced ROS is one of the crucial factors in salinity stress adaptation, and ethylene seems to play a pivotal role in ROS detoxification, thereby providing adaptation to salinity stress [35,36].

## 2. Ethylene Signal Transduction Pathway

Research in the past two decades has led to molecular dissection of the ethylene signaling network in plants [37,38,39]. In Arabidopsis, several mutants have been characterized based on the triple response of ethylene in etiolated seedlings and extensive biochemical and genetic analyses of these mutants provided the blueprint for the ethylene signal transduction pathway [40]. Ethylene is perceived by ethylene receptors, predominantly localized at the endoplasmic reticulum membrane, followed by transduction of signaling cascades to the nucleus to alter the expression of ethylene-responsive genes. In Arabidopsis, five ethylene receptors have been identified to date; including Ethylene Response1 (ETR1), ETR2, Ethylene Response Sensor1 (ERS1), ERS2, and Ethylene Insensitive4 (EIN4), which are the negative regulators of the ethylene signaling. All of the identified ethylene receptors share some common features such as the presence of three domains including an ethylene binding domain, a GAF (cGMP-specific phosphodiesterases, adenyl cyclases FhlA), and a kinase domain. Besides, some receptors also contain an additional receiver domain at their C-terminus [41]. In tomato, as many as seven ethylene receptors have been isolated to date, including LeETR1, LeETR2, NR, LeETR4, LeETR5, LeETR6 [42], and a more recently discovered SlETR7 [43]. In addition to these endoplasmic reticulum membrane-localized receptors, plasma membrane-localized ethylene receptors have also been reported. For instance, OsERS1 from rice and NTHK1 from tobacco are associated with the plasma membrane, suggesting that ethylene can be perceived at multiple locations in the cells [44].

Together with these receptors, other key components of the ethylene signaling cascade include Constitutive Triple Response1 (CTR1, a negative regulator), SIMKK (MAP kinase 4/5), MAP kinase 6, Ethylene-Insensitive2 (EIN2, a positive regulator), EIN3/EIN3-Like (EILs) (primary transcription factors), EIN3-binding F-box protein (EBF1/2), and many downstream ethylene-response factors (ERFs). Under normal physiological conditions, CTR1 remains activated upon interaction with ethylene receptors, which are the negative regulators of ethylene signaling. CTR1 is a serine/threonine-protein kinase that phosphorylates EIN2 to keep a check on the ethylene signal transduction during normal physiological conditions. Phosphorylated EIN2 is targeted by F-box proteins Ethylene Insensitive2 Targeting Protein1 (ETP1) and ETP2 for 26S proteasomal degradation [45]. Similarly, in the absence of ethylene, EIN3 is ubiquitinylated by F-box proteins Ethylene Insensitive3 Binding F-Box1 (EBF1) and EBF2 which is then targeted to ubiquitin-mediated proteolysis. Owing to the degradation of both EIN2 and EIN3, ethylene signaling is inhibited during normal growth conditions. Under stress conditions, ethylene binds to its receptors with the help of copper ions, contributed by a copper ion transporter Responsive to Antagonist1 (RAN1). Binding of ethylene to its receptors inactivates CTR1, which results in the dephosphorylation of EIN2. EIN2, in turn, activates EIN3/EILs, leading to exhaustive ethylene responses [40,45]. Some of the recent investigations reported that EIN2 is associated with EIN3 Binding F-BOX1/2 (*EBF1/2*) mRNA [46] and is also regulated by EIN2 Targeting Protein1/2 (ETP1/ETP2)-mediated protein turnover [45]. Meanwhile, *EBF1/2* mRNA targets multiple processing body (P-body) factors that stabilize EIN3/EIL1 and activates downstream events of ethylene signaling [46]. P-bodies are cytoplasmic protein complexes involved in degradation and translational arrest of mRNA.

## 3. Ethylene is a Key Modulator of Salinity Stress Responses in Plants

Ethylene biosynthesis and signaling are implicated in salinity stress tolerance in plants (Appendix A). Overproduction of endogenous ethylene or exogenous treatment of ethylene-releasing compounds such as ethephon or ethylene precursors such as 1-aminocyclopropane-1-carboxylic acid (ACC) increase salinity stress tolerance in various plants including Arabidopsis [47] and maize [12]. Moreover, ethylene has also been found as an essential positive mediator of salinity stress tolerance in grapevine [13], maize [12], and tomato [48]. Promising evidence of the involvement of melatonin in enhancing salinity stress tolerance by promoting *MYB108A*-mediated ethylene biosynthesis was also reported in grapevines [13]. It has been reported that inhibition of ethylene biosynthesis and/or signaling leads to increased sensitivity of plants to salinity stress. Gharbi et al. [48] hypothesize that ethylene exhibits a positive effect in adaptation to salinity stress probably by maintaining stomatal conductance, water use efficiency, and osmotic adjustment in *Solanum chilense* [48]. Calcium carbide (CaC_2_), a precursor of acetylene exhibiting similar effects to ethylene, is commonly used to improve seed germination rates and ethylene concentration in germinating seeds under salinity stress. It is proposed that CaC_2_ is involved in osmotic adjustments and management of oxidative stress by increasing solute concentrations and activities of antioxidant enzymes in germinating seeds. CaC_2_ considerably enhances the activity of SOD and CAT besides reducing the H_2_O_2_ and malondialdehyde (MDA) concentrations for improving seed germination in *Cucumis sativus* under salinity stress conditions [49]. These reports collectively suggest a positive regulatory role of ethylene in salt stress tolerance in plants; however, negative regulation of ethylene in salinity stress response has also been reported in many plants, including rice. Transgenic plants with reduced ethylene biosynthesis showed elevated salinity tolerance in tobacco [50]. Similarly, exogenous treatment with ethylene in rice resulted in salinity hypersensitivity [51,52]. Elevation of ethylene production under salinity stress significantly reduced growth, grain filling, and development of spikelets in rice [53]. Exogenous application of 1-MCP, an ethylene action inhibitor, to the rice spikelets resulted in improved physiological, agronomical, and biochemical characteristics under salinity stress, further suggesting a negative role of ethylene in salinity stress tolerance in rice [54].

Gene mutation and transgenic analyses have shown that almost all the components of ethylene biosynthesis and signal transduction pathway respond to salinity stress either positively or negatively. In cotton, short- and long-term salinity stress resulted in upregulation of different sets of genes of ethylene signaling involved in the regulation of salinity stress [55]. These genes include (a) ethylene biosynthesis genes (homologs of *ACS1*, *ACS12*, *ACO1*, and *ACO3*), (b) ethylene receptor genes (*ETR1*, *ETR2*, and *EIN4*), (c) ethylene signaling pathway (*ERF1*, *ERF2*, *EIN3*, and *MEKK1-MKK2-MPK4/6* kinases), and (d) feedback mechanism gene (*CTR1*) [55].

### 3.1. Salinity Stress and Ethylene Receptors

Ethylene receptors are negative regulators of ethylene signaling, and interestingly, an inhibition of ethylene receptors has been observed during salinity stress in several plant species. In Arabidopsis, salinity stress has been shown to suppress ETR1 expression [56]. In addition, *etr* loss-of-function mutants showed enhanced tolerance while *etr-1* gain-of-function mutants showed increased sensitivity to salinity stress in Arabidopsis [55,57,58]. Wilson et al. [59] reported that ETR1 and ETR2 function differently in Arabidopsis during seed germination under salinity stress. Loss-of-function mutants of *etr1* germinated earlier than the wild type (WT), while that of *etr2* germinated after WT [59]. Moreover, it has also been shown that ETR1 and ETR2 regulate abscisic acid (ABA) signaling independently of ethylene signaling and lead to contrasting germination during salinity stress [59]. In tobacco, salinity stress increases *NTHK1* mRNA levels dramatically [57], suggesting a negative regulation of ethylene in salinity stress tolerance. However, overexpression of *NTHK1* in tobacco resulted in early inductions of the ACC oxidase (*NtACO3*) and ERF (*NtERF1* and *NtERF4*) genes during salinity stress [60]. In contrast, the expression level of a salinity-inducible ACC synthase gene (*NtACS1*) was greatly suppressed in the overexpression lines. Further overexpression of *NTHK1* in Arabidopsis resulted in enhanced sensitivity to salinity stress but reduced sensitivity to ethylene [60,61]. Recently, it was reported that NTHK1 interacts with an ankyrin domain-containing protein NEIP2 (NTHK1 ethylene receptor-interacting protein 2) to improve the salinity and oxidative stress tolerance in tobacco [62]. Overexpression of *NTHK1* resulted in the accumulation of NEIP2 in the presence of both ethylene and salinity stress. On the other hand, overexpression of *NEIP2* inhibited ethylene responses, similar to the functions of NTHK1 [62]. All of these results suggest a negative regulation of ethylene receptors in salinity stress tolerance, indicating ethylene as a positive mediator of salinity stress tolerance in plants.

In addition to the ethylene receptors, CTR1 is regulated by salinity stress. It has been reported that *ctr1* loss-of-function mutants showed enhanced tolerance to salinity stress, possibly by modulation of shoot Na^+^/K^+^ ratio, which is dependent on ETR1-CTR1-regulated signaling [55,63]. Moreover, high survival rates of CTR1 mutant *ctr1-1* was observed under salinity and osmotic stress conditions in comparison to loss-of-function mutants *ein2* and *ein3-1 eil1-1* (double mutations of EIN3 and EIN3-Like1 (EIL1)) which showed remarkably reduced tolerance to salinity [58,64,65]. Ge et al. [66] showed that the heterotrimeric G-protein Gα subunit GPA1 is involved in ethylene-induced stomatal closure via NADPH oxidase-dependent H_2_O_2_ synthesis. Interestingly, GPA1 also functions downstream of RAN1, ETR1, ERS1, EIN4, and CTR1 and upstream of EIN2, EIN3, and ARR2. In guard cells of Arabidopsis leaves, ETR1 and ERS1 mediate both ethylene and H_2_O_2_ signaling, highlighting ethylene-mediated regulation of H_2_O_2_ concentrations during salinity stress [66].

### 3.2. Salinity Stress and EIN Proteins

EIN proteins are the transcription factors that function downstream of CTR1 in the ethylene signaling cascade. It has been observed that salinity stress-induced stabilization of EIN3/EIL1 promotes tolerance to salinity stress by averting ROS accumulation [55]. In Arabidopsis, loss-of-function of *ein2* resulted in enhanced sensitivity to salinity stress while its overexpression lines showed reduced sensitivity [55,65]. Lei et al. [65] identified a MA3 domain-containing protein EIN2 C-Terminus Interacting Protein 1 (ECIP1), which interacts with EIN2. *ecip1* loss-of-function mutants confer sensitivity to salinity stress during seed germination in Arabidopsis. However, these mutants show insignificant changes in ethylene responses and salinity stress tolerance. In Arabidopsis, several studies based on gene mutations showed that the ethylene signal from EIN2 to the nucleus is transduced by EIN3/EILs. Overexpression of *EIN3* remarkably enhanced tolerance to salinity stress. Interestingly, both loss-of-function mutants *ein3-1* and double mutant *ein3eil1* showed severe sensitivity to salinity stress [55,64,65]. Mao huzi (MHZ7) and MHZ6 of rice are homologs of Arabidopsis EIN2 and EIN3 [40]. OsEIL2, OsEIN2 (MHZ7), and OsEIL1 (MHZ6) were found to play a negative role in salinity stress tolerance in rice, unlike EIN2 and EIN3, which play positive roles in Arabidopsis [52]. Knockout mutants of *OsEIL2*, *MHZ7/OsEIN2*, or *MHZ6/OsEIL1* enhanced salinity stress tolerance, while their overexpression mutants exhibited increased sensitivity to salinity stress in rice [52]. Downstream of EIN3/EIL1 is *ESE1*, which is positively regulated by ethylene signaling. *ESE1* enhances plant tolerance to salinity stress by binding to promoters of salinity stress-responsive genes such as *RD29A* and *COR15A* [67]. Quan et al. [68] proposed that EIN3-SOS2 modulate salinity stress tolerance possibly by linking the ethylene signaling and salinity overly sensitive (SOS) pathways. However, further characterization showed that both EIN2 and EIN3 failed to change the expression of SOS genes in Arabidopsis. Interestingly, SOS2 phosphorylates EIN3 and activates salinity-inducible ESE1 [68]. In rice, *OsDOF15* was identified as a gene involved in the coordination between salinity and ethylene biosynthesis in rice to inhibit primary root development by affecting cell proliferation in the root apical meristem [69]. In addition, the expression levels of *OsEIL2* were found to be upregulated during salinity stress in WT plants; however, *OsEIL2* overexpression plants showed growth retardation with shortened roots and shoots than control plants [70]. In addition, *OsEIL2* overexpression plants showed increased ethylene sensitivity and accelerated leaf senescence [70]. Further, it has been observed that *OsEIL2* negatively regulates the expression of BURP genes *OsBURP14* and *OsBURP16* to reduce the pectin content during salinity stress in rice [70]. In the case of mulberry, increased expression of *MnEIL3* was observed during salinity stress both in the root and shoot [71]. Transgenic plants overexpressing *MnEIL3* showed an upregulation of ethylene biosynthetic genes in Arabidopsis to enhance salinity tolerance. Moreover, *MnEIL3* may enhance the activities of *MnACO1* and *MnACS1* promoters, indicating functioning of an ethylene–EIN3/EILs–1-aminocyclopropane-1-carboxylic acid (ACC) oxidase (ACO)/ ACC synthase (ACS) regulatory loop under salinity stress [71].

Cortical microtubule reorganization is crucial for survival under salinity stress [72,73], and ethylene has been found to modulate the same in roots cells and etiolated hypocotyls [74,75,76,77]. In Arabidopsis, microtubule-stabilizing protein Wave-Dampened2-Like5 (WDL5) takes part in ethylene signaling to inhibit etiolated hypocotyl elongation [76,78]. Dou et al. [78] found that ethylene signaling has a positive role in the regulation of microtubule reassembly and that WDL5 functions as a downstream effector of signaling involved in ethylene-mediated microtubule reassembly in salinity stress. Knockout of WDL5 partly suppressed ethylene-induced microtubule reassembly, whereas its upregulation partially protects from the salinity stress. Microtubule reassembly under salinity stress is insensitive to the effect of ACC in *ein3eil1* cells; therefore, transcriptional regulation by EIN3/EIL1 is essential for ethylene signaling-mediated reassembly of microtubules in response to salinity stress [78].

### 3.3. Effects of Salinity Stress on ERFs and other Ethylene-Responsive Transcription Factors

Ethylene-responsive element binding factors (ERFs) are transcription factors functioning downstream of EIN3 in ethylene signaling. Cheng et al. [79] reported that overexpression of *ERF1* enhances tolerance of plants to salinity, drought, and high-temperature stress conditions in an ethylene-independent manner. Moreover, enhanced expressions of three *ERF* genes including *ESE1*, *ESE2*, and *ESE3* were observed in response to salinity stress and ethylene in Arabidopsis [67]. Recently, it was shown that ERF1/2 was upregulated whereas CTR1 and EIN3-binding F-box protein 1/2 (EBF1/2) were downregulated during salinity stress in *Cynanchum auriculatum* [14]. Li et al. [80] cloned an APETALA2/ethylene responsive factor (AP2/ERF) gene from salinity-tolerant sweet potato line ND98 and named it IbRAP2-12. Based on the transient expression in tobacco epidermal cells and transcriptional activation analysis, the protein of IbRAP2-12 was found to be localized in nucleus and it was observed that IbRAP2-12 exhibits transcriptional activation because of a domain located at the C-terminus of the protein. As compared to the WT, IbRAP2-12-overexpression lines showed higher accumulation of proline and lower concentration of H_2_O_2_ under salinity stress in Arabidopsis. Moreover, multiple genes involved in the ROS-detoxification including *SAPX*, *GPX7*, and *CAT5* were found to be upregulated in the transgenic plants under salinity stress [80].

Similarly, overexpression of an ethylene-responsive transcription factor (TdSHN1) from durum wheat resulted in the development of a cuticle and lower stomatal density than WT. TdSHN1 overexpression lines, therefore, showed enhanced tolerance to salinity stress because of a reduced water loss from the leaf lamina [81]. On exposure to salinity stress, MdERF4 from apple is induced to decrease the salinity stress tolerance by binding and inhibiting the expression of *MdERF3*, suggesting that the MdERF4–MdERF3 interaction may be a feedback regulation mechanism in salinity stress to maintain ethylene homeostasis in plants [82]. Similarly, the expression of the *ERF38* gene from poplar 84K (*Populus alba* × *Populus glandulosa*) was induced under salinity stress. Overexpression of *ERF38* in poplar reduced membrane lipid peroxidation, decreased ROS accumulation, accumulated proline and soluble proteins, and exhibited higher POD and SOD activities in transgenic plants than WT, suggesting a multifaceted role of ERF38 in improving salinity and osmotic stress tolerance in poplar [83]. Acireductone dioxygenase (ARD) is an active metal-binding metalloenzyme and is involved in the formation of 2-keto-4-methylthiobutyrate (KMTB) to further produce methionine (Met) on the methionine salvage pathway, which is an initial substrate in ethylene synthesis pathway [84,85,86]. Overexpression of *OsARD1* increases the water-holding capacity and relative water content in the leaves of *OsARD1* overexpression lines to reduce the sensitivity to salinity and osmotic stresses at germination stage [87]. It was speculated that the drop in sensitivity to salinity and osmotic stresses results in an increased ethylene concentration in *OsARD1* overexpression plants [87].

## 4. Seed Germination Regulated by Ethylene under Salinity Stress

Successful seed germination is the most crucial phase in the initiation of the life cycle of plants and is regulated by many external and internal factors including phytohormones, light, temperature, drought, and salinity [88,89,90,91]. Seed germination is severely affected in saline soil, which negatively influences plant growth and crop yield [10] (Figure 1). Different components of ethylene signaling participate either positively or negatively during seed germination and seedling growth under salinity stress [92]. For example, seed germination in Arabidopsis was inhibited by ETR1 and EIN4, whereas ETR2 was found to be a positive regulator involved in stimulating seed germination during salinity stress conditions [59].

Ethylene antagonistically modulates seed germination in Arabidopsis under salinity stress via the Constitutive Photomorphogenesis1 (COP1)-mediated downregulation of Elongated Hypocotyl5 (*HY5*) and ABA Insensitive5 (*ABI5*) in the nucleus [93] (Figure 1). Salinity stress inhibits seed germination through elevation of the H_2_O_2_ and exogenous treatment of ethylene precursor (ACC) has been shown to regulate the ROS homeostasis to induce the seed germination [94]. Interaction between ethylene and nitric oxide has also been shown to regulate seed germination by decreasing the H_2_O_2_ level induced by salinity stress, further suggesting that ethylene promotes the seed germination rate by modulation of ROS production in salinity stress. On the other hand, ethylene can also inhibit the seed germination induced by salinity stress in many plant species. For example, Chang et al. [95] reported that ethylene is involved in the suppression of seed germination in cucumber (*Cucumis sativus* L.) and that l-Glu interacts with ethylene in the regulation of seed germination under salinity stress. Ethylene produced during salinity stress helps to maintain the Na^+^/K^+^ homeostasis to provide salinity stress tolerance [63].

Overexpression of the *Malus hupehensis SHINE clade protein* (*MhSHN1*) gene, a member of the AP2/ERF transcription factor, does not regulate the seed germination in transgenic tobacco plants yet it enhances salinity and osmotic stress tolerance during seed germination [96]. In *Stylosanthes humilis*, a forage legume naturally growing in the saline soils, ethylene production in the seeds provides tolerance to salinity stress [97,98,99]. Salinity induced production of ABA, and ethylene forms a point of union between the two and enables the regulation of energy metabolism and embryo growth in *S. humilis* seeds within a given pH condition [99]. Ectopic expression of a zinc finger transcription factor *Gossypium hirsutum* plant AT-rich sequence and zinc-binding (*GhPLATZ1*) in Arabidopsis regulated seed germination and seedling establishment under salinity and mannitol stress conditions. Further experimentations revealed that the inhibition of *ABI4* and *ETO1* expressions suppressed ACS gene expression to alter the ABA, GA, and ethylene pathways in transgenic lines [100]. Ectopic expression of a homolog of *AtERF38* (*GhERF38* from *G. hirsutum*) in Arabidopsis resulted in ABA sensitivity in transgenic lines; therefore, reduced seed germination under salinity and drought stress was observed in the transgenic plants as compared to WT [101]. Plant growth-promoting *Pseudomonas fluorescens* strains improve salinity tolerance of plants due to its ability to produce ACC deaminase and, consequently, to stimulate seed germination in wheat under salinity stress [102]. With the higher activity of ACC deaminase, the *Enterobacter cloacae* HSNJ4 strain could effectively promote seed germination and could provide the salinity tolerance by degrading ACC, thus inhibiting ethylene synthesis [103]. Moreover, seeds of the transgenic line overexpressing ethylene response factors (ERF95 and ERF96) showed better germination and seedling establishment as compared to the WT during salinity stress conditions [104]. A novel ethylene-responsive transcription factor from *Lycium chinense LchERF* provides salinity tolerance to transgenic tobacco during seed germination and vegetative growth [105]. Notwithstanding, Chang et al. [95] reported that salinity interferes with the ethylene signaling pathway and decreases ethylene production in seeds of *C. sativus*, which was associated with the inhibition of germination. In Faba beans (*Vicia faba*), seed germination in salinity-tolerant Y134 is not inhibited during salinity stress as compared to the salinity-sensitive Y078 probably because of the downregulation of genes related to ABA and ethylene signaling pathways and upregulation of late embryogenesis abundant (LEA) genes [106]. Seeds of *Capsicum annuum* primed with SA showed higher a germination rate due to suppression of the ethylene level as well as elevation of total soluble sugar contents and SOD activity [107].

## 5. Fine-Tuning of Photosynthetic Machinery by Ethylene during Salinity Stress

Homeostasis of essential elements like N, P, K, S, and Ca is altered during salinity stress, which in turn affects the photosynthetic efficiency of plants [108,109,110]. Salinity stress induces oxidative stress through increased production of ROS, which can disrupt chloroplast functions (Figure 2). Salts at higher concentrations induce both osmotic and ionic stresses, which affect photosynthetic activity either by closing the stomata or by reducing the activity of CO_2_-fixing enzymes and availability of water in the plant cells [111] (Figure 2). Activities of CO_2_-fixing enzymes are reduced at higher concentrations of Na^+^, and tolerance of these enzymes to Na^+^ concentrations varies from species to species [112]. Na^+^ ions imbalance the proton motive force and thus influence photosynthetic machinery and chloroplastic functions [113]. Salinity stress influences the photosynthetic parameters including chlorophyll, photosystems, net photosynthesis rate (Pn), chlorophyll fluorescence parameters, soluble sugar contents, and ribulose bisphosphate carboxylase/oxygenase (RuBisCO) activity [114]. Recently, it was reported that tomato plants showed improved photosynthesis, metabolic homeostasis, and growth rate as a result of elevated CO_2_ under salinity stress by decreasing the amount of ABA hormone and ACC [115]. Among all the photosynthetic parameters, photosystem II (PSII) is the most susceptible to various abiotic stresses including salinity [116,117]. Homeostasis of Na^+^ ions maintain membrane integrity, relative water content, net photosynthesis, and yield. Ethylene has been shown to promote the homeostasis of Na^+^/K^+^, nutrients, and ROS to enhance plant tolerance to salinity [40]. The *ctr1-1* mutants maintain relatively higher concentrations of K^+^ and lower concentrations of Na^+^ in contrast to *ein2-5* or *ein3* plants, where an opposite trend of K^+^ and Na^+^ concentration was observed compared with the WT undertreated and optimum conditions [63]. Because of this altered K^+^ and Na^+^ homeostasis, *ctr1-1* plants displayed a slight reduction in leaf area and root elongation, while *ein2-5* or *ein3-1* mutants showed magnified retardation in plant growth compared to the WT under salinity stress [63]. In pomegranate, salinity decreased the net photosynthetic rate, chlorophyll content, stomatal conductance, relative water content, and electrical conductivity [114,118,119]. Further heat map analysis showed that antiapoptotic genes *BAG6* and *BAG7* were clustered together with *ERS2*, *EIN3*, and *ACS* and that the transcripts levels of *BAG6*, *BAG7*, *ERS2*, and *ACS2* were significantly suppressed in the response to salinity. The inclusion of ACC or ethylene source in the saline solution restored the expression levels of *BAG6* and *BAG7*, suggesting the involvement of ethylene in the regulation of these antiapoptotic genes under salinity stress [120].

In Arabidopsis and alfalfa, *Enterobacter sp.* SA187 mediates salinity tolerance by producing 2-keto-4-methylthiobutyric acid (KMBA), which is converted into ethylene *in planta* [121,122]. This *Enterobacter*-produced KMBA is involved in the maintenance of photosynthesis and primary metabolism together with the reduction of ABA-mediated stress responses in plants. Gene expression analysis revealed that, after SA187 inoculation, genes related to photosynthesis and primary metabolism remain unaltered under salinity stress conditions as compared to the mock plants [122,123]. Similarly, in rice, inoculation of *Glutamicibacter* sp. YD01 facilitated rice plants to combat stress by ethylene-mediated regulation of ROS accumulation, ion homeostasis, and photosynthetic capacity and by enhancing stress-responsive gene expression [124]. Tight regulation of ROS homeostasis also accelerates photosynthesis and growth by abating lipid peroxidation in chloroplasts [125,126,127].

It is well established that salinity stress also affects nitrogen and sulfur assimilation in plants. Plants grown on low nitrate (5 mM) showed lower photosynthesis and growth compared to the plants grown in sufficient nitrate concentrations during non-saline conditions [128]. When excess nitrate (20 mM) was applied under non-saline conditions, an inhibitory effect on photosynthesis was observed, which was related to higher ethylene production. However, under salinity stress conditions, as the demand for N increased, the excess N optimized ethylene, enhanced proline production, and promoted photosynthesis and growth [128]. Recently, it was reported that cadmium and sodium stress conditions induce ethylene and Jasmonic acid (JA) signaling. Both of these signaling pathways converged at *EIN3/EIL1* and resulted in an enhanced expression of a nitrate transporter *NRT1*.8 and reduced expression of *NRT1*.5. Although it resulted in decreased plant growth, it promoted plant tolerance to stress in a nitrate reductase-dependent manner by mediating the stress-initiated nitrate allocation to roots, which decoupled nitrate assimilation and photosynthesis [129]. Taken together, these studies clearly highlight that ethylene plays a major role in stabilizing photosynthesis under salinity stress conditions by maintaining the ROS accumulation, ion homeostasis, and mineral homeostasis and by elevating the antioxidant defense mechanism.

## 6. Modulation of Plant Cell Death under Salinity Stress

Several hormones including jasmonic acid, salicylic acid, abscisic acid, gibberellins, and ethylene regulate plant cell death (PCD). Of these, ethylene has been most actively involved in PCD and is alone sufficient to cause PCD in many plant systems [130]. PCD is a vital process during growth and development, especially in the organization and maintenance of plants, and plays a crucial role in the several defense responses against certain types of environmental stresses, both abiotic including salinity stress [131,132] and biotic ones [133,134,135,136,137]. ROS accumulation is an important inducer of PCD in plant cells under stress conditions as ROS molecules such as H_2_O_2_ and superoxide radicals (O_2_^−^) can damage the cellular components [138]. Similarly, salinity stress-induced ROS also initiates PCD during salinity stress conditions [139]. In tomato suspension cells, ethylene production enhances PCD during salinity stress [140]. Ethylene also regulates the expression and activity of cysteine proteases, an important component of the NaCl-induced cell death program [141]. It has been reported that cysteine protease inhibitor E-64 decreases NaCl-induced cell death. During abiotic stress conditions and senescence [142,143], cysteine protease subfamily C1A (papain-like cysteine proteases) components cause PCD [144]. The expression of the cysteine protease gene in *Petunia* corollas is regulated by ethylene sensitivity. The expression of genes encoding the two cysteine protease inhibitors *AtCYSa* and *AtCYSb* improves salinity stress tolerance in Arabidopsis [145]. Cell death signaling can be mediated by the activation of MAPKs (Mitogen-activated protein kinase) and cysteine proteases, and ethylene can speed-up the process only in cells exposed to high salinity [140,146]. Moreover, decreased concentrations of K^+^ and the ratio of K^+^/Na^+^ have been reported to be involved in the cell death caused by high salinity [147]. Silver thiosulfate, an ethylene receptor blocker, significantly reduces the NaCl-induced cell death, confirming the role of ethylene in direct control of PCD [140].

Ethylene responsive transcription factor *ERF109* gene retards PCD and improves salinity tolerance in the WT tobacco plant [148]. The role of ERFs in the adaptation to biotic and abiotic stresses is well established [149,150]. It was observed that the *ERF109* gene has a positive role in salinity-induced lateral root development [148]. Moreover, it was also reported that ethylene can control the salinity-induced PCD and can enhance salinity stress tolerance in Arabidopsis by maintaining ROS homeostasis [120].

*Bcl-2*-associated athanogene (BAG) family genes are also associated with PCD during growth, development, and stress responses. In Arabidopsis, ethylene stimulates the expression of senescence-associated genes (*ANACO29*, *ANACO92*, *RPK*, *SAG*, *ATG GBF*, and *GDH*) [151] and ethylene-related genes (*ERS2*, *ERF1*, *ACS2*, *ETR2*, and *EIN3*) [152]. After comparison of BAG gene expression levels with ethylene-related and senescence-associated genes in the WT under salinity stress as well as after adding ACC, it was observed that salinity alone was able to significantly suppress the transcript levels of *BAG6* and *BAG7*. However, exogenous application of ACC in the saline solution restored the expression levels of *BAG6* and *BAG7* and inhibited salinity-induced PCD. Ethylene and salinity antagonistically regulate BAG family-, ethylene-, and senescence-related genes to mediate salinity-induced PCD and to confer salinity tolerance in Arabidopsis [120].

## 7. Cross-Talk between Ethylene and Various Plant Hormones during Salinity Stress

Phytohormones like auxin (IAA), cytokinin (CK), and abscisic acid (ABA) interact with ethylene to play a decisive role in adaptation to the salinity stress [153] (Figure 3). In response to environmental fluctuations, calcium-dependent protein kinases (CDPK) and MAPKs signaling cascades are stimulated in parallel and their partial overlap helps in exhibiting stimulus-specific response. It is ethylene which is responsible for mediating cross-talk between these two pathways and for exhibiting the desired response [154]. Furthermore, there exists an antagonistic interaction between ethylene and gibberellins, which helps in adjusting salinity tolerance either by regulating or initiating the defense system [155] (Figure 3).

### 7.1. Auxin and Ethylene

There exists a cross-talk between phytohormones to overcome the adverse environmental conditions faced by a plant (Figure 3). The cross-talk between IAA and ethylene results in stimulation of the antioxidation mechanism. It was observed that *diageotropica* (*dgt*), a tomato mutant plant with reduced IAA sensitivity, was able to avoid cadmium stress due to a disruption in IAA-triggered ethylene biosynthesis [156]. Moreover, IAA-stimulated ethylene production controls ABA; therefore, plants grew better even under abiotic stress [157].

### 7.2. Cytokinin and Ethylene

CK is the hormone associated with growth and development. It promotes cell division, shoot differentiation, leaf senescence, and other developmental processes. After its synthesis in the root tips, it is transported to the shoots. It was reported that synthesis and transportation of CK are reduced in response to salinity stress, leading to lower salinity tolerance [158]. In the same study, antagonistic interaction between ABA and CK was also revealed during salinity stress, which hinders the tolerance. External application of CK to the salinity-stressed plants resulted in downregulation of the genes involved in ABA-mediated stress response [159]. Similarly, lower expressions of mitogen activated protein kinase kinase 9 (*MKK9*), ACC synthase 6 (*ACS6*), and *ERF1* and *ERF5* were observed after external application of CK, which were upregulated before CK application under salinity stress, further suggesting the antagonistic interaction of CK with ABA and ethylene (Figure 3).

### 7.3. Gibberellins (GA) and Ethyslene

Under stress conditions, GA interacts with ethylene via aspartate-glutamate-leucine-leucine-alanine (DELLA) proteins [160]. Ethylene increases the accumulation of DELLA proteins, which in turn reduce the action of gibberellins (Figure 3). Plants with enhanced accumulation of DELLA proteins showed better tolerance to salinity stress as compared to the plants where the accumulation of DELLA proteins was not observed [64]. Moreover, GA, in order to tolerate salinity stress, may act downstream of ethylene signaling. The interaction of GA with ethylene was observed via DELLA proteins only. Moreover, the mutants in which the DELLA protein gene was disrupted by four folds showed improved root growth in high salinity stress, as ethylene suppresses the root growth and GA promotes the same [161].

### 7.4. Abscisic Acid and Ethylene

ABA, well-known for its involvement in stomata closure, also plays crucial roles during salinity stress tolerance. An increase in ABA concentration was observed in the leaves of tomato plant as a response to salinity. Salinity stress leads to water scarcity in plants due to low soil water content and high vapor pressure due to climatic conditions. This water deficiency stimulates the closure of stomata via ABA to prevent water loss by transpiration [162]. ABA and salinity are known to regulate a number of genes associated with ethylene biosynthesis and signaling including *AtACS5*, *AtACS7*, *TaACO1*, and *GhERF1*, among others [40]. EIN2 is the key regulator and mediator between ethylene and other phytohormones, such as ABA, and alteration in the EIN2 during salinity stress results in an increased amount of ABA production [163]. Mutations in ACS7 lead to an upregulation of stress-responsive genes involved in ABA signaling, thereby enhancing the plant’s tolerance to salinity stress [164]. Moreover, reduced ethylene concentration and leaf abscission were observed in citrus plants subjected to external application of ABA under salinity stress [165].

### 7.5. Jasmonic Acid, Brassinosteroids, and Ethylene

Jasmonic acid (JA), a lipid-derived plant hormone, plays essential roles in plant growth and withstanding harsh environmental conditions [166]. JA acts as a positive controller of salinity stress tolerance in plants [167]. A study aimed to analyze the effect of vermicompost leachate (VCL) in salinity-stressed tomato seedlings showed that, though the concentration of JA in VCL was extremely low, its endogenous concentration in roots and leaves was significantly higher. This can only be attributed to VLC-stimulated JA synthesis [168]. Exogenous application of JA helps in inducing salinity tolerance in soybean [169], tomato [170], and rice [171]. JA is also known to improve photosynthetic activity [158] and to reduce Na^+^ concentration under salinity stress. Ethylene and JA may either work together or antagonistically in response to stress conditions. The cross-talk between JA and ethylene is mediated via EIN3/EIL1 along with JAZs-MYC2 [172]. JA and ethylene act synergistically and repress leaf growth and expansion by targeting AUXs which are responsible for growth and suppressor of JA synthesis. Gene expression is affected via JA2 repressor and *EIN3/EILI* transcription factor as ethylene promotes the stability of EIN3/EILI [154]. In addition, a synergistic action of these two phytohormones was observed during stress conditions through the activation of ERF1. In addition, it has also been shown that *EIN3*-deficient mutant showed an enhanced concentration of JA, further suggesting a cross-talk between these two hormones via EIN3 [173].

Brassinosteroids (BRs) and ethylene interact with each other by initiating an alternative respiratory pathway (Figure 3). Exogenous application of BRs to cucumber seedlings showed increased levels of ethylene and alternative oxidase pathway (AOX) components. BRs also help in avoiding oxidative damage done by ROS as it initiates the production of ethylene and ROS, which in turn activates AOX, leading to the detoxification of excess ROS produced [174]. BR and ethylene act synergistically in overcoming the adverse effects of salinity during seed germination. Salinity stress hinders proper seed germination by affecting ACO synthesis [175]. A study on tomato plants highlighted a relation among H_2_O_2_, ethylene, and BR [176]. Ethylene was involved in the BR-induced salinity tolerance, and H_2_O_2_ was responsible for enhanced BR-induced ethylene accumulation. Moreover, inhibition of H_2_O_2_ synthesis also reduced BR-induced ethylene accumulation during salinity stress. Similarly, when ethylene was inhibited, H_2_O_2_ generation was partially blocked. Taken together, these results highlight a vital role of ethylene and H_2_O_2_ in BR-induced salinity stress tolerance.

## 8. Conclusions

This review summarizes recent advancements highlighting the involvement of ethylene in salinity stress tolerance in plants (Appendix A). Research carried out to date on ethylene and salinity stress confirms that in planta ethylene levels may positively or negatively affect plants’ responses to salinity stress, suggesting that the fine-tuning of ethylene action may be necessary for salinity stress tolerance in plants. Ethylene also interacts with other phytohormones to bring about the desired response, either by regulating the gene expression or by regulating transcription factors. Although a growing body of evidences suggests a cross-talk of ethylene with other phytohormones during salinity stress, a detailed mechanism is still elusive and needs to be investigated. Based on the current understanding of the roles of ethylene under salinity stress, a putative model is proposed (Figure 4). Future efforts could be on the further dissection of ethylene-induced signaling during salinity stress through a multi-omics approach. Moreover, efforts should also be made in the future to investigate the roles of posttranslational modifications in the regulation of ethylene signaling under salinity stress conditions. The quest for the biological role of ethylene under salinity stress must lure plant biologists to pursue the investigation with solid experimental evidence.

## Figures and Tables

**Figure 1 biomolecules-10-00959-f001:**
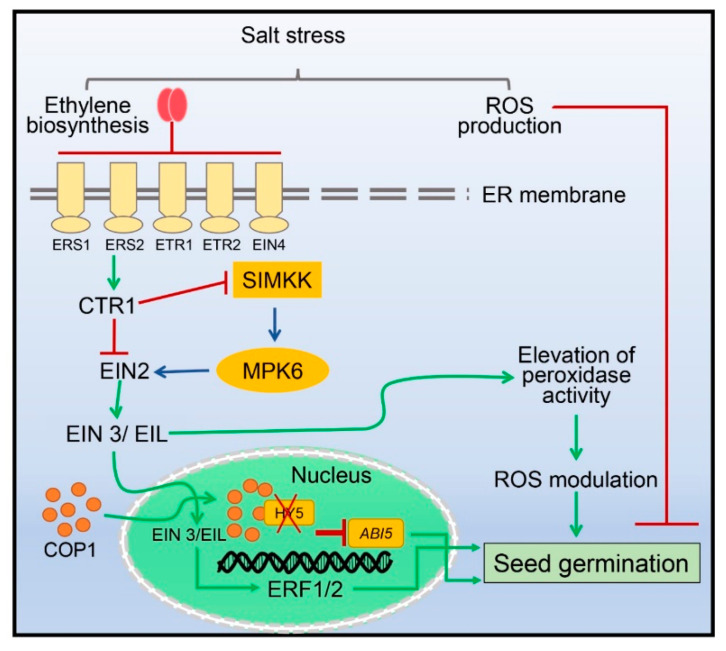
Role of ethylene signaling in seed germination under salinity stress. Ethylene induced by salinity stress activates the signaling pathway by inhibiting the active receptors and by releasing the Constitutive Triple Response1 (CTR1). Salinity-induced ethylene signal is transduced mainly through the classical receptors–CTR1-Ethylene-Insensitive2 (EIN2)-EIN3 pathway to regulate many effectors involved in plant growth and salinity response. EIN3/EIN3-Like (EIL) promotes the entry of Constitutive Photomorphogenesis1 (COP1) into the nucleus and degrades the Elongated Hypocotyl5 (HY5) protein, which inhibits seed germination by upregulating *ABI5* gene expression. Degradation of HY5 inhibits the expression of *ABI5* and ultimately induced seed germination under salinity stress. On the other hand, EIN3/EIL also directly induces seed germination by scavenging reactive oxygen species (ROS) through upregulating peroxidase activities.

**Figure 2 biomolecules-10-00959-f002:**
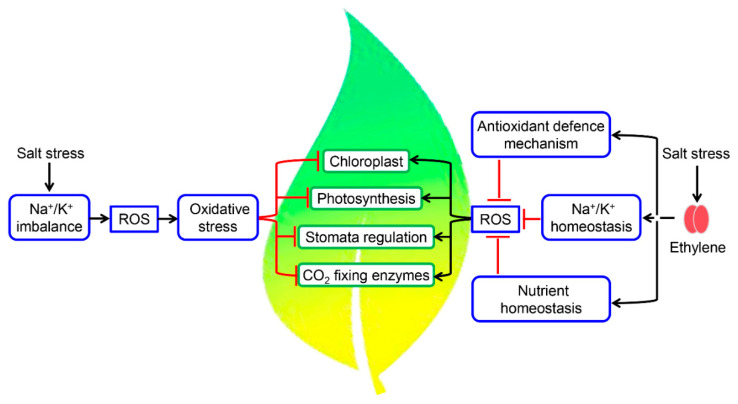
Functions of ethylene in the regulation of photosynthesis under salinity stress. In the absence of ethylene, salinity stress results in an imbalance of Na^+^/K^+^ homeostasis, which leads to the production of ROS. This salinity-induced ROS production, in turn, exerts oxidative stress on plants, resulting in stomatal closure and reduced activity of CO_2_-fixing enzymes, resulting in a decrease in photosynthesis. In the presence of ethylene, Na^+^/K^+^ homeostasis and nutrients homeostasis are maintained, and the antioxidant defense mechanism is activated, which limits ROS production, thereby preventing ROS-induced oxidative stress. In the absence of oxidative stress, the rate of photosynthesis is maintained even during salinity stress.

**Figure 3 biomolecules-10-00959-f003:**
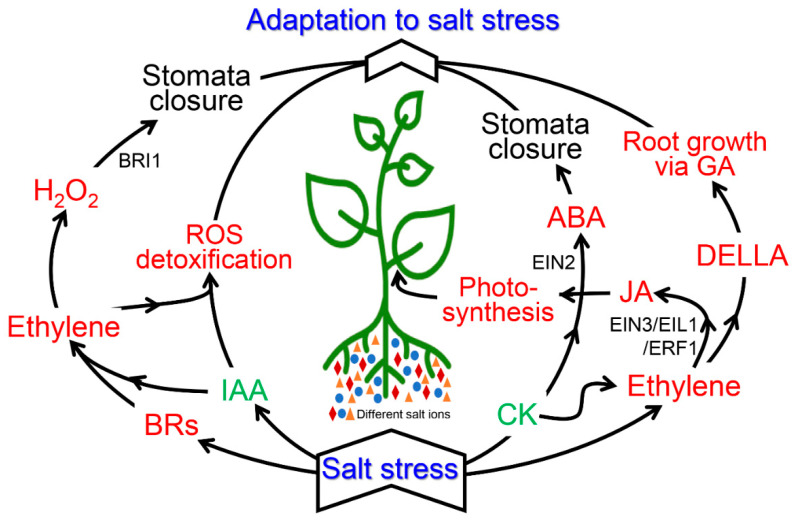
Schematic diagram of hormonal cross-talk under salinity stress. Details of the cross-talk are given in the text. Abbreviations used: BR: Brassinosteroids, IAA: Indole-acetic acid. CK: Cytokinin, ABA: Abscisic acid, GA: Gibberellin, JA: Jasmonic acid.

**Figure 4 biomolecules-10-00959-f004:**
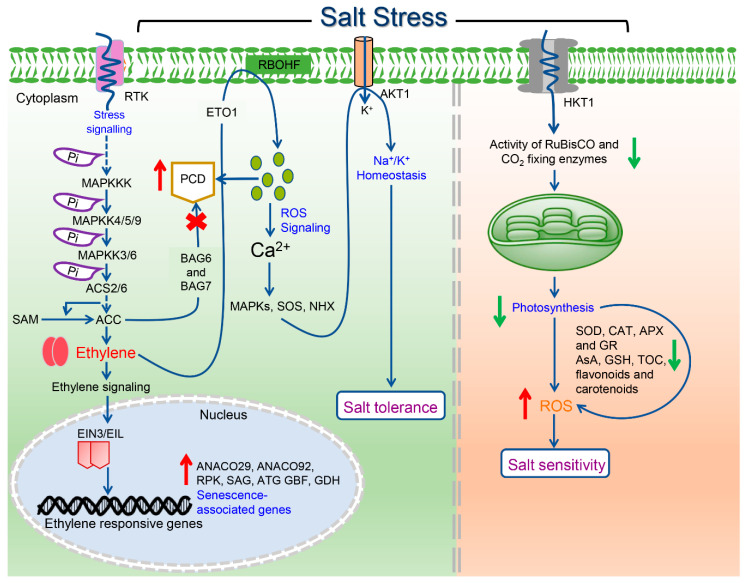
A schematic representation of ethylene regulation under salinity stress. When plants encounter ionic imbalance, nutrient imbalance, and osmotic stress under salinity stress, activation of upstream mitogen activated protein kinase kinase kinases (MAPKKKs) leads to the gradual phosphorylation and successive activation of downstream mitogen activated protein kinase kinases (MAPKKs) and MAPKs that further phosphorylates ACS2/6. The phosphorylated ACS2/6 stabilizes and thus enhances the production of ethylene through ethylene biosynthesis. Consequently, the downstream signaling activates the transcription of several ethylene-responsive genes. The transcription level of BAG (*Bcl-2*-associated athanogene) family genes is suppressed by ethylene (1-aminocyclopropane-1-carboxylic acid (ACC)) to inhibit salinity-induced plant cell death (PCD). On the other hand, ethylene enhances ROS signaling to increase calcium ions, which in turn regulate MAPK, salinity overly sensitive (SOS), and Na^+^/H^+^ exchangers (NHX) through Na^+^/K^+^ homeostasis to provide salinity stress tolerance. Salinity stress reduces the CO_2_-fixing enzymes in chloroplast that eventually produce ROS. Scavenging of ROS occurs through both enzymatic (superoxide dismutase (SOD), catalase (CAT), ascorbate peroxidase (APX), and glutathione reductase (GR)) and nonenzymatic (ascorbic acid (AsA), glutathione (GSH), α-tocopherol (TOC), flavonoids, and carotenoids) antioxidant defense systems.

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
