# Peer review of "Ethylene: A Master Regulator of Salinity Stress Tolerance in Plants"

_biomolecules, 2020, doi:10.3390/biom10060959_

Round 1

Reviewer 1 Report

Dear Authors,

Reviewer comments biomolecules-837361

The review manuscript entitled „Ethylene: A master regulator of salinity stress tolerance in plants“ represents a valuable overview of recent studies dealing with a role of ethylene in plant responses to salinity stress. The manuscript provides basic information on ethylene receptors, signalling pathways, ethylene-responsive transcription factors, as well as recent results on ethylene involvement in regulation of physiological and developmental processes such as seed germination, photosynthesis or programmed cell death (PCD) as affected by salinity stress. In addition, cross-talks of ethylene with other plant hormones including auxin, cytokinins, giberellins, abscisic acid, jasmonic acid and brassinosteroids in salt-treated plants are discussed. The manuscript text is accompanied by four figures providing schematic diagrams summarising recent knowledge on ethylene role in signaling and seed germination, regulation of photosynthesis, hormonal crosstalk with other hormones, and a summarising scheme on salt stress effects on ethylene signaling in plants.

I think that the manuscript is worth publishing.

I have only one suggestion how to improve the manuscript and several formal comments which should be corrected prior to the manuscript publication.

1/ I would recommend the authors to add a supplementary table providing an overview of the recent studies on the role of ethylene in salt-treated plants including basic information on plant material used (such as mutants in ethylene metabolism or signaling), experiment arrangement, and obtained results as a supplementary material to the manuscript.

2/ Formal comments on the text:

Abstract, line 22: Correct the typing error in the verb „has focused“ (not „focussed“).

Abstract, line 28: Modify the sentnece as follows: „…to elevating the assimilation of nitrates and sulfates.“

Introduction, line 46 and further in the text: and a semicolon and a comma preceding and folowing the word „therefore“ in the sentence „…and, therefore, ethylene production predominantly takes place locally at the site of its action…“

Introduction, line 47: Add the word „studies“ following the words „Some of the earlier“, i.e., „Some of the earlier studies showed that…“

Introduction, line 83: Be concise in using terms and thus use the word „adaptation“ instead of „adaption“ in „salinity stress adaptation“ since the word „adaptation“ is used on the following line („providing adaptation to salinity stress“).

Line 130: Remove „a“ preceding the words „promising evidence“ since „evidence“ is an uncountable noun.

Line 138: Remove „the“ preceding the term „osmotic adjustments“ in the sentence „It is proposed that CaCl2 is involved in osmotic adjustments and management of oxidative stress…“

Line 144: Modify the sentence as follows: „It has been shown that transgenic plants with downregulated ethylene biosynthesis showed elevated salinity tolerance…“

Line 159: Add a comma both preceding and following the word „interestingly“ and use a singular form of the verb „has been observed“ in the sentence „Ethylene receptors are the negative regulators of the ethylene signaling and, interestingly, an inhibition of ethylene receptors has been observed during salinity stress in several plants.“

Line 187 and further in the text: I think that the reference should be cited as „Ge et al. [68]“ instead of „Ge et al., (2015)“ but the authors should check the journal´s instructions.

Line 219: Add a semicolon and a comma preceding and following the word „however“ in the sentence „…during salinity stress in WT plants; however, OsEIL2 overexpression plants showed growth retardation…“

Line 225: Modify the word form „upregulated“ to „an upregulation“ in the sentence „Transgenic plants overexpressing MnEIL3 showed an upregulation of ethylene biosynthetic genes in Arabidopsis…“

Line 238: Add a semicolon and a comma preceding and following the word „therefore“ in the sentence „Microtubule reassembly under salinity stress is insensitive to the effect of ACC in ein3eil1 cells; therefore, transcriptional regulation by EIN3/EIL1 is supposed to be essential for ethylene signaling…“

Line 249: Add the word „it“ following the word „named“ in the sentence „Li et al., (2019) cloned an APETALA2/ethylene responsive factor (AP2/ERF) gene from salinity-tolerant sweet potato line ND98 and named it IbRAP-12…“

Line 308: The current version of the sentence is very long and it should be divided into two shorter ones, the second one starting with „and ultimately modification of ABA, GA and ethylene pathways…“

Line 319: Add a comma both preceding and following the word ůconsequently“ and modify the word form „stimulating“ to „to stimulate“ in the sentence „Plant growth-promoting Pseudomonas fluorescens strains improve salinity tolerance of plants due to its ability to produce ACC deaminase and, consequently, to stimulate seed germination…“

Line 350: Add the either word „at“ or „under“ in the sentence „Activities of CO2 fixing enzymes are reduced at (under) higher concentrations of Na+…“

Line 364: Use eitehr the term „control conditions“ or „optimum conditions“ as an opposite to salinity stress instead of using „normal conditions“.

Line 371: Replace the word „of“ by „to“ in the words „in the response to salinity.“

Line 386: Add a space between a number and a corresponding unit in „5 mM“

Line 390: Use the word „enhanced“ instead of „higehr“ in „…and led to the enhanced proline production…“

Line 392: Add the word „pathways“ following the word ůsignaling“ in the sentnece „Both of these signaling pathways..“

Line 394: Remove „a“ preceding the words „decreased plant growth“ in the sentnece „Although it resulted in decreased plant growth,..“

Line 406: Add the word „ones“ following the words „…against certain types of Environmental stresses, both abiotic including salinity stress.. and biotic ones…“

Line 434: Correct the evrb form „was able to suppress“ (not „was able to suppressed“) and use the word forms „transcript levels“ (not „transcripts level“) in the sentence „…it was observed that salinity alone was able to significantly suppress the transcript levels of BAG6, BAG7.“

Line 439: Replace the word „different“ by „other“ in the heading „7. Cross talk of ethylene with other plant hormones during salinity stress conditions“.

Line 440: Change the word order „abscisic acid (ABA)“ (not ůabscisic (ABA) acid“).

Line 457: Add a semicolon and a comma preceding and following the word „therefore“ in the sentence „Moreoevr, IAA-stimulated ethylene controls ABA; therefore, plants grew better even under abiotic stress..“

Line 515: Add the word „components“ at the end of the sentnece „…cucumber seedlings showed increased levels of ethylene and alternative oxidase pathway (AOX) components.“

Line 526: Figure 4 legend: Modify the words „Schematically representation“ to „A schematic representation of ethylene regulation..“

Final recommendation: Accept after a minor revision.

Author Response

The review manuscript entitled „Ethylene: A master regulator of salinity stress tolerance in plants“represents a valuable overview of recent studies dealing with a role of ethylene in plant responses to salinity stress. The manuscript provides basic information on ethylene receptors, signalling pathways, ethylene-responsive transcription factors, as well as recent results on ethylene involvement in regulation of physiological and developmental processes such as seed germination, photosynthesis or programmed cell death (PCD) as affected by salinity stress. In addition, cross-talks of ethylene with other plant hormones including auxin, cytokinins, giberellins, abscisic acid, jasmonic acid and brassinosteroids in salt-treated plants are discussed. The manuscript text is accompanied by four figures providing schematic diagrams summarising recent knowledge on ethylene role in signaling and seed germination, regulation of photosynthesis, hormonal crosstalk with other hormones, and a summarising scheme on salt stress effects on ethylene signaling in plants.

I think that the manuscript is worth publishing.

Reply: Thank you for your positive and constructive comments on our manuscript.

  1. I would recommend the authors to add a supplementary table providing an overview of the recent studies on the role of ethylene in salt-treated plants including basic information on plant material used (such as mutants in ethylene metabolism or signaling), experiment arrangement, and obtained results as a supplementary material to the manuscript.

-Reply: Thank you for the suggestion. We have added the table (Table S1) as per the suggestion.

  1. Formal comments on the text:
  2. Abstract, line 22: Correct the typing error in the verb „has focused“ (not „focussed“).

Changed to: Research in the past three decades, therefore, has focused on analyzing the effects of salinity stress on the plants.

  1. Abstract, line 28: Modify the sentence as follows: „…to elevating the assimilation of nitrates and sulfates.“

Changed to: Ethylene modulates salinity stress responses largely via maintaining the homeostasis of Na+/K+, nutrients, and ROS by inducing antioxidant defense in addition to elevating the assimilation of nitrates and sulfates.

  1. Introduction, line 46 and further in the text: and a semicolon and a comma preceding and folowing the word „therefore“ in the sentence „…and, therefore, ethylene production predominantly takes place locally at the site of its action…“

Changed to: Because of the gaseous nature of ethylene, it can easily diffuse to nearby cells and, therefore, ethylene production predominantly takes place locally at the site of its action.

  1. Introduction, line 47: Add the word „studies“ following the words „Some of the earlier“, i.e., „Some of the earlier studies showed that…“

Changed to: Some of the earlier work showed that ethylene biosynthesis is dramatically induced during fruit ripening and leaf senescence, among others [4].

  1. Introduction, line 83: Be concise in using terms and thus use the word „adaptation“ instead of „adaption“ in „salinity stress adaptation“ since the word „adaptation“ is used on the following line („providing adaptation to salinity stress“).

Changed to: Successful detoxification of the stress-induced ROS is one of the crucial factors in salinity stress adaptation and ethylene seems to play a crucial role in ROS detoxification, thereby providing adaptation to salinity stress [35,36].

  1. Line 130: Remove „a“ preceding the words „promising evidence“ since „evidence“ is an uncountable noun.

Changed to: Promising evidence of the involvement of melatonin in enhancing salinity stress tolerance by promoting MYB108A-mediated ethylene biosynthesis was also reported in grapevines [13].

  1. Line 138: Remove „the“ preceding the term „osmotic adjustments“ in the sentence „It is proposed that CaC2 is involved in osmotic adjustments and management of oxidative stress…“

Changed to: It is proposed that CaC2 is involved in osmotic adjustments and management of oxidative stress by increasing solute concentrations and activities of antioxidant enzymes in germinating seeds.

  1. Line 144: Modify the sentence as follows: „It has been shown that transgenic plants with downregulated ethylene biosynthesis showed elevated salinity tolerance…“

Changed to: It has been shown that transgenic plants with downregulated ethylene biosynthesis showed elevated salinity tolerance in tobacco [52].

  1. Line 159: Add a comma both preceding and following the word „interestingly“ and use a singular form of the verb „has been observed“ in the sentence „Ethylene receptors are the negative regulators of the ethylene signaling and, interestingly, an inhibition of ethylene receptors has been observed during salinity stress in several plants.“

Changed to: Ethylene receptors are the negative regulators of the ethylene signaling and, interestingly, an inhibition of ethylene receptors has been observed during salinity stress in several plants species.

  1. Line 187 and further in the text: I think that the reference should be cited as „Ge et al. [68]“ instead of „Ge et al., (2015)“ but the authors should check the journal´s instructions.

Changed to: Ge et. al. [68], showed that the heterotrimeric G-protein Gα subunit GPA1 is involved in ethylene-induced stomatal closure via NADPH oxidase-dependent H2O2 synthesis [68].

  1. Line 219: Add a semicolon and a comma preceding and following the word „however“ in the sentence „…during salinity stress in WT plants; however, OsEIL2 overexpression plants showed growth retardation…“

Changed to: In addition, the expression levels of OsEIL2 were found to be upregulated during salinity stress in WT plants; however, OsEIL2 overexpression plants showed growth retardation with shortened roots and shoots than control plants [72].

  1. Line 225: Modify the word form „upregulated“ to „an upregulation“ in the sentence „Transgenic plants overexpressing MnEIL3 showed an upregulation of ethylene biosynthetic genes in Arabidopsis…“

Changed to: Transgenic plants overexpressing MnEIL3 showed an upregulation of ethylene biosynthetic genes in Arabidopsis to enhance salinity tolerance.

  1. Line 238: Add a semicolon and a comma preceding and following the word „therefore“ in the sentence „Microtubule reassembly under salinity stress is insensitive to the effect of ACC in ein3eil1 cells; therefore, transcriptional regulation by EIN3/EIL1 is supposed to be essential for ethylene signaling…“

Changed to: Microtubule reassembly under salinity stress is insensitive to the effect of ACC in ein3eil1 cells; therefore, transcriptional regulation by EIN3/EIL1 is essential for ethylene signaling mediated reassembly of microtubules in response to salinity stress [80].

  1. Line 249: Add the word „it“ following the word „named“ in the sentence „Li et al., (2019) cloned an APETALA2/ethylene responsive factor (AP2/ERF) gene from salinity-tolerant sweet potato line ND98 and named it IbRAP-12…“

Changed to: Li et. al. [82] cloned an APETALA2/ethylene responsive factor (AP2/ERF) gene from salinity-tolerant sweet potato line ND98 and named it IbRAP2-12 [82].

  1. Line 308: The current version of the sentence is very long and it should be divided into two shorter ones, the second one starting with „and ultimately modification of ABA, GA and ethylene pathways…“

Changed to: Ectopic expression of a zinc finger transcription factor Gossypium hirsutum plant AT-rich sequence and zinc-binding (GhPLATZ1) in Arabidopsis regulated seed germination and seedling establishment under salinity and mannitol stress conditions. Further experimentations revealed the inhibition of ABI4 and ETO1 expressions suppressed ACS gene expression to alter the ABA, GA and ethylene pathways in transgenic lines

  1. Line 319: Add a comma both preceding and following the word ůconsequently“ and modify the word form „stimulating“ to „to stimulate“ in the sentence „Plant growth-promoting Pseudomonas fluorescens strains improve salinity tolerance of plants due to its ability to produce ACC deaminase and, consequently, to stimulate seed germination…“

Changed to: Plant growth-promoting Pseudomonas fluorescens strains improve salinity tolerance of plants due to its ability to produce ACC deaminase and, consequently, to stimulate seed germination in wheat under salinity stress [104].

  1. Line 350: Add the either word „at“ or „under“ in the sentence „Activities of CO2 fixing enzymes are reduced at (under) higher concentrations of Na+…“

Changed to: Activities of CO2 fixing enzymes are reduced at higher concentrations of Na+ and tolerance of these enzymes to Na+ concentrations varies from species to species [114].

  1. Line 364: Use eitehr the term „control conditions“ or „optimum conditions“ as an opposite to salinity stress instead of using „normal conditions“.

Changed to: The ctr1-1 mutants maintain relatively higher concentrations of K+ and lower concentrations of Na+ in contrast to ein2-5 or ein3 plants where an opposite trend of K+ and Na+ concentration was observed compared with the WT under-treated and optimum conditions [65].

  1. Line 371: Replace the word „of“ by „to“ in the words „in the response to salinity.“

Changed to: Further heat map analysis showed that anti-apoptotic genes BAG6, BAG7 were clustered together with ERS2, EIN3, and ACS, and the transcripts level of BAG6, BAG7, ERS2, and ACS2 were significantly suppressed in the response to salinity.

  1. Line 386: Add a space between a number and a corresponding unit in „5 mM“

Changed to: It is well established that salinity stress also affects nitrogen and sulfur assimilation in plants. Plants grown on low nitrate (5 mM) showed lower photosynthesis and growth compared to the plants grown in sufficient nitrate concentrations during non-saline conditions [130].

  1. Line 390: Use the word „enhanced“ instead of „higehr“ in „…and led to the enhanced proline production…“

Changed to: However, under salinity stress conditions, as the demand for N increased, the excess N optimized ethylene and led to the enhanced proline production and promoted photosynthesis and growth [130].

  1. Line 392: Add the word „pathways“ following the word ůsignaling“ in the sentnece „Both of these signaling pathways..“

Changed to: Both of these signaling pathways converged at EIN3/EIL1 and resulted in an enhanced expression of a nitrate transporter NRT1.8 and reduced expression of NRT1.5.

  1. Line 394: Remove „a“ preceding the words „decreased plant growth“ in the sentnece „Although it resulted in decreased plant growth,..“

Changed to: Although it resulted in decreased plant growth, it promoted plant tolerance to stress in a nitrate reductase- dependent manner by mediating the stress-initiated nitrate allocation to roots, which decoupled nitrate assimilation and photosynthesis [131].

  1. Line 406: Add the word „ones“ following the words „…against certain types of Environmental stresses, both abiotic including salinity stress.. and biotic ones…“

Changed to: PCD is a vital process during growth and development, especially in the organization and maintenance of plants and also plays a crucial role in the several defense responses against certain types of environmental stresses, both abiotic including salinity stress [133,134] and biotic ones [135-139].

  1. Line 434: Correct the verb form „was able to suppress“ (not „was able to suppressed“) and use the word forms „transcript levels“ (not „transcripts level“) in the sentence „…it was observed that salinity alone was able to significantly suppress the transcript levels of BAG6, BAG7.“

Changed to: it was observed that salinity alone was able to significantly suppress the transcript levels of BAG6, BAG7.

  1. Line 439: Replace the word „different“ by „other“ in the heading „7. Cross talk of ethylene with other plant hormones during salinity stress conditions“.

Changed to: Cross talk between ethylene and various plant hormones during salinity stress

  1. Line 440: Change the word order „abscisic acid (ABA)“ (not ůabscisic (ABA) acid“).

Changed to: Phytohormones like auxin (IAA), cytokinin (CK), and abscisic acid (ABA)  interact with ethylene to play a decisive role in adaptation to the salinity stress [155] (Figure 3).

  1. Line 457: Add a semicolon and a comma preceding and following the word „therefore“ in the sentence „Moreoevr, IAA-stimulated ethylene controls ABA; therefore, plants grew better even under abiotic stress..“

Changed to: Moreover, IAA-stimulated ethylene production controls ABA; therefore, plants grew better even under abiotic stress [159].

  1. Line 515: Add the word „components“ at the end of the sentnece „…cucumber seedlings showed increased levels of ethylene and alternative oxidase pathway (AOX) components.“

Changed to: Exogenous application of BRs to cucumber seedlings showed increased levels of ethylene and alternative oxidase pathway (AOX) components.

  1. Line 526: Figure 4 legend: Modify the words „Schematically representation“ to „A schematic representation of ethylene regulation..“

Changed to: A schematically representation of ethylene regulation under salinity stress.

Reviewer 2 Report

I have reviewed the manuscript and find it to be a significant contribution to the field of research pertaining to signaling in salinity stress as it pertains to ethylene.

I found several grammatical errors/writing style in the manuscript, which I cannot list here but have highlighted them in the attached annotated pdf file for the benefit of the authors. I am happy for the manuscript to be accepted after incorporation of these corrections and any other grammar issues I might have missed.

Author Response

I have reviewed the manuscript and find it to be a significant contribution to the field of research pertaining to signaling in salinity stress as it pertains to ethylene.

I found several grammatical errors/writing style in the manuscript, which I cannot list here but have highlighted them in the attached annotated pdf file for the benefit of the authors. I am happy for the manuscript to be accepted after incorporation of these corrections and any other grammar issues I might have missed.

-Reply: Thank you for your positive and constructive comments on our manuscript. We have incorporated the changes as per the suggestion wherever required.